# Von Hildebrand on the Roots of Moral Evil

**Martin Cajthaml** 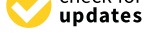

Sts Cyril and Methodius Faculty of Theology, Palacky University, CZ-779 00 Olomouc, Czech Republic; martin.cajthaml@upol.cz

**Abstract:** In this article, I sketch, both in broad outlines and in selected details, the new, richer picture of von Hildebrand's account of moral evil as it emerges from my discovery of extensive materials in von Hildebrand´s *Nachlass* at the Bavarian State Library in Munich dealing with the "roots of moral evil". These manuscripts and typescripts, the critical edition of which will be published at the same time as this article or shortly thereafter, show that von Hildebrand´s account of moral evil is much richer, more nuanced, and complex than the one we can glean from the final section of *Ethics*, his magnum opus in moral philosophy. In this article, I also aim to situate von Hildebrand´s analysis of the roots of moral evil in the context of both Christian religious thought and the Western philosophical tradition. Von Hildebrand was, to be sure, an heir to both of these traditions, despite the thrust of his phenomenological method to "bracket" all extant theories and turn "back to the things themselves". The mind-boggling feature of the tension between von Hildebrand´s existential rootedness in the Catholic tradition and his methodological distance to it, including the Aristotelian–Thomist philosophy, is the following: On one hand, he claims that the two ultimate roots of moral evil are pride and concupiscence, which sounds perfectly traditionally Christian. On the other hand, however, he strips these concepts of most of their traditional connotations and endows them with the meaning they acquire in the context of his phenomenological analyses. The intriguing result of this approach is the transformation of religious or moral theological concepts of pride and concupiscence into descriptive phenomenological categories which encompass an almost inexhaustible wealth of various subspecies and subordinate forms of moral evil.

**Keywords:** moral evil; akrasia; value; pride; concupiscence; hatred; realist phenomenology; Christian ethics

## 1. Introduction

Dietrich von Hildebrand was a former student of Edmund Husserl´s and an important figure in early phenomenology. Besides his seminal religious writings, he made notable contributions to several branches of philosophy, including epistemology, aesthetics, philosophical anthropology, philosophy of love, and social philosophy (von Hildebrand 1975, 1991, 2007, 2009, 2016). However, he is most renowned and appreciated as a moral philosopher (von Hildebrand 1969a, 1969b, 1980, 2017, 2019a, 2019b, 2020). His magnum opus in this field, *Ethics*, which was first published almost seventy years ago, recently reappeared in a third and definitive edition (von Hildebrand 2020). Among the many themes treated in this voluminous, rich, and exacting work, there is also an intriguing treatment of the origins of moral evil. It appears right at the end of the book and, until quite recently, it seemed to be the most extensive treatment of this topic in von Hildebrand's oeuvre. However, this misleading impression was recently rectified by my discovery of extensive materials in von Hildebrand's *Nachlass* at the Bavarian State Library in Munich dealing with the "roots of moral evil".[1] These manuscripts and typescripts, just about to be published by the Hildebrand Press, show that von Hildebrand's account of moral evil is much richer and more nuanced and complex than the one we can glean from the final section of his *Ethics* (von Hildebrand 2023).[2]

In this article, I present von Hildebrand's account of moral evil as it emerges from both of these principal sources, that is, *Ethics* and the edited materials. Writing a paper on this topic is justified not just because the edited materials bring a new, richer perspective on the theme, but also because even the account of moral evil found in *Ethics* has not yet been discussed in the scholarly literature. Hence, the contribution of the present article is twofold. First, it inaugurates the scholarly treatment of von Hildebrand's analysis of the roots of moral evil. Second, it does so by drawing on primary sources which were discovered and edited only quite recently and will come out shortly before or after the publication of this article.

## 2. The Points of Contact of von Hildebrand's Account of Evil with the Tradition

The aim of this section is to mark some points of contact between von Hildebrand's account of moral evil and the traditional accounts of the same subject in Western philosophical and religious thought. This is important because von Hildebrand himself, in the spirit of Husserl's phenomenological maxim, "Back to the things themselves", largely abstains from a discussion with his predecessors. His declared aim is to describe and scrutinize the "given", that is, the moral experience as such (von Hildebrand 2020, p. 2). But it is obvious that despite this effort to "bracket" all extant theories in order to focus on the "moral data", von Hildebrand's phenomenological analysis is not performed in a conceptual vacuum. Moreover, a brief reminder of some traditional views about the origins of moral evil will prove to be helpful in assessing the originality of von Hildebrand's analysis.

One notable point of contact between von Hildebrand's account of evil and the traditional views is the problem of akrasia or acting against one's better knowledge.[3] In ch. 3 of *Ethics*, one of the most important texts in his whole ethical oeuvre, von Hildebrand takes issue with the Socratic claim that "No one errs knowingly", meaning that the knowledge of the best is a sufficient precondition for acting morally well. Von Hildebrand argues that if I am in a situation where I have to choose between, say, participating in an amusing social event and assisting my distressed friend, I may choose the former option, knowing well that the latter option is, morally speaking, better (von Hildebrand 2020, p. 44). In situations such as this, he contends, I am usually not confronted with a choice between a higher and a lesser good, with the implication that there is a single scale of values on which both of these goods are to be found. Rather, I am confronted with one course of action which appeals to me from the point of view of pure subjective satisfaction and another course of action which appeals to me from the point of view of its intrinsic value; to amuse oneself is merely subjectively satisfying, and to help one's friend in distress is valuable in itself.

Von Hildebrand elaborates on the contrast between what is merely subjectively satisfying and what is important in itself in the immediately preceding section of ch. 3 of *Ethics* (von Hildebrand 2020, pp. 36–41). He points out that while what is important in itself is the intrinsic value of some being, the merely subjectively satisfying is an entirely subjective type of importance that persons, things, or events acquire in the human experience as a result of causing subjective satisfaction or dissatisfaction in us. This total divorce in principle of the merely subjectively satisfying as a category of motivation from the objective structure of a given being makes it incommensurable with the motivation rooted in the intrinsic worth of things and persons around us. This brings von Hildebrand to say that there is no "common denominator" between these two "categories of importance" (von Hildebrand 2020, pp. 42–44). That means, for example, that by intensifying the mere subjective satisfaction, we are not coming closer to the viewpoint of the intrinsically important. It also means that it is impossible to consider the merely subjectively satisfying as a lower type of intrinsic importance.

It is on the basis of this insight that von Hildebrand explains how acting against one's better knowledge is possible. If I choose to amuse myself rather than to assist my friend in distress, it is because I chose the merely subjectively satisfying over the intrinsically important, knowing that, from the moral point of view, doing what is intrinsically important is unequivocally better than doing just what is pleasing. This type of explanation of

acratic action is essentially different from the one given by the tradition. Both Plato and Aristotle explain acratic action not by a morally reprehensible choice/volition, but by strong emotions or desires interfering with one's capacity for practical reasoning.[4] Ever since Augustine, the decisive factor in explaining the possibility of acting against one's better knowledge has been the will. But the explanation of how such willing is possible was given in terms of the distinction between the true and apparent good. I choose what is worse by preferring what appears to me to be the higher good, while, in fact, it is lower. Max Scheler's approach is also similar. In his view, I prefer a value, which, objectively speaking, is lower.[5] This is possible as a result of my disordered ordo amoris, which makes me love more what is of lesser value and love less what is of higher value.[6]

These accounts are fundamentally flawed as explanations of the dilemma of the amusing event vs. assisting one's friend in distress and other such situations as they explain away the actual choice of what one knows to be, morally speaking, worse. They interpret the situation as one in which someone acts on his or her (culpable or inculpable) ignorance of what, in the given situation, is the higher good. Thus, instead of explaining how it is possible to go with one's will against one's knowledge of the best, they collapse, ultimately, into a restatement of the old Socratic dictum that "No one errs knowingly", with the qualification that strong emotions or desires may, at times, overturn the dominion of the cognitive element in our psyche over the appetitive one with the result of swaying us in the direction immanent in the desire or in the emotion.

Compare this type of explanation of acratic action to von Hildebrand's. In his view, my choice to go to the amusing event instead of assisting my friend in distress is not based on my (culpable) ignorance of what is the higher good in the given situation or, in Scheler's terminology, which value is preferred. When making my decision, I know well that, from the viewpoint of what is intrinsically important, it is clearly better to assist my friend in distress than to go to the amusing event. I also know that, from the perspective of mere subjective satisfaction, going to the amusing event is definitely preferable to the other alternative. So, the moral decision here is ultimately a choice between the following two incommensurable viewpoints: the appeal of the intrinsically important vs. the attraction of the merely subjectively satisfying. This explanation makes it fully understandable why our decisions (and actions based on them) are sometimes at odds with our knowledge of the best. It also explains how hesitation is possible in such situations. For, seen exclusively from one of the two viewpoints, no hesitation is possible. It is only possible when the two viewpoints clash. And they clash because each of the alternatives appeals to me from one of the two perspectives.

It may seem that, given von Hildebrand's account of akrasia, the main root of moral evil must be the striving for the merely subjectively satisfying. However, von Hildebrand contends that striving for the merely subjectively satisfying is, as such, not morally bad (von Hildebrand 2023). Therefore, it cannot be considered the ultimate root of moral evil. The ultimate root of moral evil in the human person, he submits, is her interest in the merely subjectively satisfying, *which is detached from the value-response attitude* (von Hildebrand 2023).[7] In order to understand this important qualification, it must be explained what the value-response attitude is for von Hildebrand. In the discussion of akrasia, we encountered the value-response attitude in the form of a value response to a specific value in a particular situation (helping a friend in need).[8] However, according to von Hildebrand, value responses given in particular situations of our lives flow organically from value responses which "are not restricted to a mere actual experience but must subsist in a superactual way if they are at all real and not mere sham acts". (von Hildebrand 2020, p. 253). Von Hildebrand calls these value responses *superactual* (*überaktuell*). Their most characteristic example is love. Love does not cease to exist the moment it ceases to be currently experienced, when, for example, we stop thinking about or talking to a loved one and devote ourselves to something else, while, for example, a pain in the back that recurs repeatedly may have the same physiological causes, yet it will not be one and the same pain that persists in us even in moments when we are not currently experiencing

it; we can say that love for a certain person persists in us even in moments when our current consciousness is focused elsewhere and is present in the background of our current consciousness and gives it a certain coloring.

Unlike love, which is always directed at a specific person with its unique and unrepeatable value, some superactual value responses are directed at whole value spheres. For example, if we say that someone is not interested in art, we are not expressing his or her attitude toward this or that work of art, but toward artistic values in general. Von Hildebrand calls these superactual responses *general* (*allgemein*). This term does not indicate that the attitudes or responses themselves are general or universal. They are the concrete and real attitudes or responses of a concrete and real person. It is their *object* that is general. For it is, as noted above, not a particular value or good, but a type of value or a sphere of goods. These attitudes are also general in the sense that they give rise to attitudes toward the particular values and goods to which the general response relates, that is, for example, the general attitude toward artistic values is the basis of the responses to particular works of art and their value given by the person who has this general attitude. This general superactual response always has its basis in a certain sensibility of the particular person for the sphere of goods or type of values to which the answer refers. The generality of this response can vary. For example, one may be sensitive to music, but not to the whole field of art. Someone else may be sensitive to the whole field of art, etc. The broader the range of goods and types of values to which one is sensitive, the more general one´s attitude or response will be. Usually, more general attitudes fund less general attitudes. According to von Hildebrand, by analyzing these increasingly general attitudes, we arrive at what he calls fundamental or basic attitudes.

Turning now to the moral sphere, the "backbone" of moral virtue is, for von Hildebrand, always a certain *general superactual value response*. Von Hildebrand thus develops a singular account of the nature of moral virtue that is distinct from the Aristotelian notion of moral virtue as a habit, and the mean between two extremes (Cajthaml 2012). In *The Art of Living*, he depicts the following five basic closely related moral virtues: reverence, faithfulness, responsibility, veracity, and goodness (von Hildebrand 2017). These fundamental moral attitudes are particular expressions of the value-response attitude of which von Hildebrand speaks when he says that the ultimate root of moral evil is the interest in what is merely subjectively satisfying that is detached from the value-responding attitude. To put it in the traditional moral language, von Hildebrand says that the ultimate source of moral evil is the pursuit of pleasure unbridled by moral virtue. When put in this way, von Hildebrand´s complex phenomenological terminology dissolves into a very familiar statement. Yet, how is such a detachment of the pursuit of pleasure from moral virtue possible? And which forms can it assume? The answer to these questions brings us to the very heart of von Hildebrand's understanding of the ultimate roots of moral evil.

Von Hildebrand's answer to the first question is that there are opposing tendencies in human nature. This answer in itself is nothing new. We already find it in Plato's theory of the tripartite soul in *Republic* IV and in innumerable later accounts of the human psyche or, say, in later philosophers' reflections on the ambivalence of the human will. What is original, however, is the way in which von Hildebrand describes these opposing tendencies. He first does so in the last chapters of *Ethics* and then in the edited materials. In ch. 31 of *Ethics*, he describes three "centers" in the human person: the free, loving, value-responding center, and the two opposing "centers" of pride and concupiscence. On this basis, he develops, in ch. 32, an intriguing account of five basic types of morally deficient characters. It is based on the observation that although the value-responding center in the human person is antithetical to the centers of pride and concupiscence, it nevertheless coexists with them in most human beings in several basic ways. Sometimes this coexistence has the character of an open fight between the antithetical centers, as in an upright but still morally struggling person. Sometimes the value-responding center is merely juxtaposed with the centers of pride and concupiscence, as in the morally unconscious person. Sometimes, again, a compromise is struck between the antithetical centers, as in the third type, etc.[9]

A notable corollary to this account of opposite "centers" in the human person is von Hildebrand's contention that moral good and evil are *polar opposites*. By this claim, von Hildebrand seems to contradict the millennia-old tradition according to which evil is only a privation of the good.[10] However, one must note that von Hildebrand does not dispute the metaphysical doctrine of good and evil and explicitly limits his claim to the relationship between *moral* good and evil. According to his account, moral evil is the polar opposite of good, because evil human attitudes, such as hatred, are not merely a lack of morally good attitudes, such as love. The lack of love is indifference, not hatred, von Hildebrand says. Moreover, from an epistemological point of view, hatred has its own intelligible structure that can be investigated in its own right and cannot be conceived as a sort of "imperfect benevolence".[11] The nature, forms, and relations of this attitude to other phenomena, say, pride, are open to philosophical inquiry and can be investigated in their own right, without bringing in the presupposition of the metaphysical meaning of evil as privation.

In this context, it is easy to imagine the following objection to von Hildebrand. Suppose he tries to stay on the plane of moral good and evil. Can he avoid the ontological plane altogether? Does a certain Manicheanism not manifest itself in the insistence that human nature has, in addition to a positive tendency toward moral good (a value-response center), a negative tendency toward moral evil (pride, concupiscence)? What is the ontological status of these opposing "centers"? Are they all ontologically on the same level?

Von Hildebrand is aware of this possible objection and tries to answer it at the very end of his discussion of the roots of moral evil in the edited materials. The following passage deserves to be quoted in full:

> "This analysis of the roots of moral evil has disclosed to us that pride and concupiscence are always at the basis of all moral evil. The metaphysical question arises: where do these two morally negative centers enter into the human person? It is obvious that they do not come from God, that they are not something issuing from God's hand in creating man. Every creature of God is positive, possessing value, reflecting in some way His infinite goodness. 'Nothing is evil, but the perversion of our will,' says St. Augustine.
>
> Are they a result of the fall of man? Many symptoms of concupiscence and pride are certainly the sad heritage of original sin, as the tendency of our nature to leave the attitude of *religio* when we are confronted with the subjectively satisfying, the rebellion of our instincts against our spirit, the immanent logic of our nature: the continuous tendency of our nature to infect our good intentions by pride and many other symptoms of the mysterious rupture and disharmony in our fallen nature that, notwithstanding its negative character, has such a tremendous reality.
>
> But is not the original sin due to pride and concupiscence? One may answer: Pride and concupiscence are but a privation and nothing positively existing. This may be true, but it does not explain the mystery. The question arises: Where does this privation come from?
>
> We do not pretend to be able to answer this question and to explain the mysterious temptation of pride—potentially connected to the priceless privilege of free will. We restrict ourselves to stating the two following fundamental facts. Firstly, God can never be the cause of pride and concupiscence. Secondly, pride and concupiscence exist in the fallen man and are the roots of all moral evil." (von Hildebrand 2023)

It is clear from this important passage that von Hildebrand leaves open the question of the metaphysical status of negative "centers". He admits that they could be a mere privation of the due good. But he points to the fact that *even if* we consider them only as a lack of goodness, we have in no way answered the crucial question, that is, what the actual cause of this "lack" is. Von Hildebrand thus suggests that the decisive question, namely, what the final cause of all evil is, is unanswerable on the philosophical level. Hence, even the so-called privation theory of evil cannot ultimately explain the mystery of evil.

Now that I have indicated the relation of von Hildebrand's conception of the roots of moral evil to tradition (akrasia and the privation theory of evil), I will now turn to this conception itself. In his account, von Hildebrand, as already mentioned, focuses on the analysis and description of concupiscence and pride as the two main roots of moral evil and on the analysis of hatred as one of its main manifestations or consequences.[12]

## 3. Concupiscence

In von Hildebrand's view, the basic difference between pride and concupiscence is that while concupiscence delves into and throws itself at subjectively satisfying goods, the proud person is characterized by a reflexive gaze at herself (von Hildebrand 2020, p. 465). The relationship between subjective gratification and ego (self) thus differs significantly in each case. To use von Hildebrand's Old Testament metaphor, in concupiscence, "one renounces one's birthright for a mess of pottage" (von Hildebrand 2020, p. 465). In pride, the person arrogates to herself a right that surpasses her; she exalts herself in an illegitimate way. Despite this general distinction between pride and concupiscence, von Hildebrand is very careful not to speak of pride and concupiscence in general. He distinguishes three basic types of concupiscence, more precisely, three basic types of persons afflicted with concupiscence, and a total of four basic types of pride.

The first type of concupiscent person is one in whom "the craving for the agreeable assumes a violent form" and whose "temperament has an impetuous character" (von Hildebrand 2020, p. 462). This is the best-known type of concupiscent person, personified, say, by Don Giovanni from the famous Mozart opera or by the father of the Karamazov brothers in Dostoyevsky's novel. The second is "the vegetative, phlegmatic type in whom concupiscence has the character of a lazy and heavy enslavement to the agreeable" (von Hildebrand 2020, p. 463). "Men in this category", writes von Hildebrand, "do not manifest any passionate impetus, any unquenchable thirst which holds them in a state of perpetual tension. Comfort plays a greater role in their life than the intensely agreeable. Theirs is a bovine heaviness characterized by the predominance of the desire not to be disturbed in the satisfaction of their animal urges. They are too lazy for any passionate craving for the agreeable. But they are nevertheless exclusively absorbed by the subjectively satisfying, mostly concerned with bodily agreeable things, but they also enjoy in a 'Fafnerlike' manner the possession of wealth. Their charnel, blunt approach toward the world makes them completely indifferent to the reign of morally relevant values and incapable of being charitable. They share hardness with the first type, but their hardness has more the character of a blunt and pachyderm-like insensitivity" (von Hildebrand 2020, p. 463).

The third and "soft" type is characterized neither by the passionate craving for the agreeable of the first nor the bovine enslavement by the agreeable of the second. The soft type "reacts in an exaggerated, self-pitying manner to any bodily displeasure or pain" and "likes to be petted, to be surrounded by a soft atmosphere, to be caressed and cherished" (von Hildebrand 2020, p. 463). "He is the man", von Hildebrand writes, "who pities himself, who feels himself to be harshly treated on every occasion, who is incapable of abandonment to the important-in-itself and of any interest in it for its own sake. He always thinks of himself and his own feelings, and relishes every emotion, instead of focusing on the object which motivates this emotion. His tears are rooted only in his pampered softness and brought on exclusively by the slightest harsh or rough touch; sometimes he even enjoys them for their own sake" (von Hildebrand 2020, p. 463).

The first two types, which von Hildebrand calls "hard", are, despite the quite obvious differences between them, actually shown on several occasions to be much closer to each other than to the "soft" type. What the three types do share in common, however, is total egocentrism, with the only interest in life being the gratification of the desire for the agreeable. As von Hildebrand notes, this fundamental trait makes all of them equally incapable of charity (von Hildebrand 2020, p. 464).

In addition to these three basic ways in which a person can be affected by concupiscence, von Hildebrand makes a number of other distinctions in the sphere of concupiscence. The most basic of these, from a systematic point of view, is the distinction between the pursuit of mere subjective satisfaction, which may or may not have the morally negative character of concupiscence, and the pursuit of mere subjective satisfaction, which is inherently illegitimate and, thus, always a manifestation of concupiscence. He calls the latter "pure" concupiscence. What is remarkable is his observation that "pure" concupiscence has two forms, sadism and curiosity (in the sense of the sensational indiscretion with which some people inquire into the intimate affairs of other people). Now, these two are so different that, at first sight, it would seem that they have nothing in common at all. But this is not the case according to von Hildebrand's analysis. It reveals that the common feature of both of these otherwise quite different phenomena is a desire for mere subjective satisfaction, which is inherently reprehensible and, therefore, cannot take any legitimate form. Unlike, say, sexual gratification, which can take morally legitimate and morally illegitimate forms, the desire of the sadist and the sensationalist is always wrong. The pleasure caused by the sight of another's pain and suffering, or the pleasure the sadist derives from being able to inflict suffering on another person, is always morally reprehensible, regardless of circumstances, historical and cultural conditions and contexts, etc. The same is true, according to von Hildebrand, of indiscriminate sensationalism.

Nonetheless, despite this basic similarity between sadism and sensationalism, there is, von Hildebrand contends, something which sets the former apart from the latter and, in fact, from all types of concupiscence. It is the fact that the source of evil in sadism is not just the disrespect for morally relevant values, discarding the basic value-responding attitude. The sadist's desire for pleasure is different to all other types of desire in that it is necessarily connected with the evil character of the object itself—not with its formal character of being evil as such, but with its material nature, which is intrinsically morally evil. What gratifies this immoral desire is the fact that one knowingly and willingly inflicts suffering upon another person. To take pleasure in such an activity is morally evil, even if the perpetrator believes it is just punishment for the wrongdoings of the victim; it is even more obviously morally evil if the one who suffers is innocent (von Hildebrand 2023).

The distinction between "pure" concupiscence and the pursuit of pleasure, which may or may not be a form of concupiscence, opens up the very complex question of what causes a legitimate form of the pursuit of pleasure (or avoidance of pain) to become morally illegitimate. Von Hildebrand addresses this question by discussing various forms of pleasure seeking and the avoidance of pain. In the course of these analyses, he makes several intriguing points. One of them is the following asymmetry in the concern with physical pleasure and pain; while the pursuit of physical pleasure is always accompanied by the danger that one will abandon the value-responding attitude and that one's relation to the pleasure in question will thereby be stained by concupiscence, the effort of avoiding physical pain does not carry this intrinsic danger.

Here, von Hildebrand points out an interesting fact that has not been noticed in traditional treatises on pleasure, as far as I know. The pursuit of pleasure, he claims, is in itself morally neutral. It becomes morally legitimate only if certain conditions are fulfilled. By contrast, the avoidance of pain and suffering is in itself morally positive. The effort to avoid pain and suffering becomes morally bad only if there is some special obligation to undergo the pain and suffering in question. For example, if someone wants to force me to bear false witness by means of the threat of torture, I may have an obligation to undergo the pain of torture if it cannot be avoided except at the cost of a morally wrong act. By contrast, the pursuit of pleasure might be immoral even if there is no special obligation to abstain from it. But if this is the case, under what conditions is the pursuit of pleasure morally positive?

In von Hildebrand's view, the morality of our desire for pleasure depends on whether this desire is subordinated to the above-mentioned basic value-response attitude. In this context, he very interestingly points out the inadequacy of the concept of moderation,

which is based on the idea of the autonomy of the will in the face of the allure of pleasure (von Hildebrand 2023). He recognizes that this autonomy is definitely an important formal prerequisite for leading a morally good life, as otherwise, despite having the best intentions, one falls prey to one's passions. However, what is crucial from a moral point of view is not the ability to control one's desires by one's will, but the dominion of the aforementioned basic value-response attitude over the centers of pride and concupiscence. It is this preponderance that enables us to find a morally proper approach to pleasure. By contrast, the ability of our will to renounce certain pleasures can be instrumentalized to achieve other, more refined pleasures. If we look at a person dominated by the desire for primitive pleasure and a person dominated by the desire for a more sophisticated pleasure from the perspective of the concept of moderation just discussed, the latter would appear to be morally superior to the former because a higher degree of moderation can be presupposed. However, such a perspective is misleading from a moral point of view, von Hildebrand suggests. A person who is dominated, for example, by the desire for power, and who sacrifices many of the pleasures of a comfortable life to satisfy that desire is not morally superior in that respect. On the contrary, a person may be morally worse than a primitive hedonist despite the fact that she displays a superior capacity for moderation in comparison with the primitive hedonist.

In the edited materials, von Hildebrand does not limit himself to the description and analysis of concupiscence in relation to bodily pleasures and pains. He also turns to the area of psychological pleasure and pain. This passage is perhaps even more interesting than the discussion of bodily pleasures and displeasures. In it, von Hildebrand accumulates insights and analyses approaching what we are used to finding in the existentialist philosophy and in the literature. He explores such various phenomena as games, superficial forms of socializing, the light literature, movies, pop music, and even the concupiscence, which can be detected in the satisfaction of one's psychic urges and in the unloading of one's mental energies.

Below, I limit myself to just two notable points that appear in this section. Both of them concern concupiscence in the area of superficial socializing. Within this area, von Hildebrand distinguishes several phenomena. One of them is the concupiscence of the "elegant man" who seeks the superficial, elegant surroundings of a salon. The tendency of this type of person to substitute a basic value-response attitude for conventional forms of politeness and elegance is, von Hildebrand aptly notes, mostly unconscious and hidden under a mask of good manners. Yet, even here, the ultimate roots of this substitution are pride and concupiscence. Concupiscence, observes von Hildebrand, manifests itself in the desire for superficial distraction that is characteristic of these forms of socializing, in the yearning for "the illusion of a world without suffering and serious problems; the soft atmosphere of politeness and compliments" (von Hildebrand 2023). Pride shows itself in the desire to satisfy this concupiscence while avoiding the humiliating experience of losing self-control, which would tarnish the illusion of the grand social image this "gentleman" has created of himself. According to this fine analysis, this "gentleman's" pride hides his concupiscence from him. Notes of this type testify to von Hildebrand's remarkable capacity for subtle psychological analysis, which, unlike the authors of psychological novels such as Dostoyevsky, is manifested not in the description of individual human characters, but in the description of certain more general character types. In many cases, however, it is not difficult for the reader to find examples of the character described by a literary character or an actually existing person. Sometimes it is von Hildebrand himself who furnishes such fitting examples.

The other particularly notable point in von Hildebrand's description of concupiscence in the area of superficial socializing is his observation that the preference of illusion to reality is a result of concupiscence. He makes this observation in the context of the discussion of the person who pursues superficial socializing in the hope of forgetting, or of escaping the burdensome awareness of her own troubles. Von Hildebrand argues that the tendency of such a person to flee from oppressive reality contradicts the value-response attitude. What

is interesting about this observation is the finding that concupiscence is not only an attitude in which one strives for subjective satisfaction, but also an attitude for which the object, its meaning, and its positive or negative value are not essential. The only thing that matters to such an attitude is subjective states (von Hildebrand 2023). The following passage is worth quoting in full:

> "To prefer an illusion to reality is already a result of concupiscence. The respect for reality, the desire to face it, to remain in conformity with reality and truth is an important formal element of *religio* and the fundamental value-response attitude. The attitude in which one tries to do away with something depressing by ignoring it, by lulling oneself into the illusion that it is not so, in attempting to forget it in plunging into superficial distractions and pleasures, lacks respect for reality and reveals a formal subjectivism which is a fruit of concupiscence.

> Additionally, the trend to fly into the periphery, to escape the cross in stepping down to a lower level, to render ourselves in a lower stratum of our life, to yield to the easier way, to avoid the inner effort and *elan*, is a typical fruit of concupiscence. The deep metaphysical laziness which incites us to choose the easier way, to fear any actualization of our depth, is rooted in concupiscence, as is all laziness. And the readiness to give up the deeper strata, to fly into the periphery, which means as such an ignoring of the general *sursum corda* of the reign of the values, an indifference toward their call, is equally rooted in concupiscence". (von Hildebrand 2023)

As mentioned in this quotation, the tendency to escape the weight of reality is not only manifested in the search for superficial entertainment and company. It is also manifested by laziness. Von Hildebrand develops this theme further in a short typescript on this topic, which is included in the edition. There, he distinguishes between laziness in the sense of an unwillingness to work, whether manually or intellectually, and spiritual laziness. And he observes that even "the most assiduous man, efficient in his work, endowed with a great potential for activity and agility, whom we would contrast with the inert and lazy man, the man who could not live without work, who relishes business, who is reliable, punctual, self-controlled, may be typically lazy in the sense of spiritual laziness". (von Hildebrand 2023). This remark is worthy of careful reflection in today's world, which is dominated by work, efficiency, and pragmatism as an existential attitude.

## 4. Pride

Following this rich, complex, yet systematically elaborate account of concupiscence, von Hildebrand turns to the other main root of moral evil, which is pride. Like concupiscence, pride has many forms. In ch. 35 of his *Ethics*, von Hildebrand distinguishes four basic types of pride. The first, also the rarest and most radical, is satanic pride. This pride aspires to "metaphysical lordship" or "metaphysical grandeur" (von Hildebrand 2020, p. 466). It is because of this aspiration that it revolts in hatred against every value. "The fundamental gesture of this pride", von Hildebrand submits, "is an impotent attempt to dethrone all values, to deprive them of their mysterious metaphysical power" (von Hildebrand 2020, p. 466).

The second basic form of pride is the "pride of self-glorification". It is directed at the values in oneself; however, it does not aim at "dethroning" them. Rather, a person dominated by this form of pride uses values as the means for self-glorification, as the "source of one's grandeur" (von Hildebrand 2020, p. 468). In the edited material, the pride of self-glorification is further divided into sub-types, each of which relates to the set of values that is exploited (von Hildebrand 2023). The most morally abominable is self-glorification in religious and moral values. Von Hildebrand calls it the pride of the Pharisee. A less serious type is pride taken in one's intellectual values. Even less objectionable is taking pride in one's good looks, bodily strength, and so on. Hence, the moral seriousness

of this type of pride depends on the position of the respective values in the hierarchy of values.

The third basic type of pride is vanity (von Hildebrand 2020, p. 471). It consists of relishing the possession of real or alleged perfections. Vanity does not feature the immoral attitude to the values we find in both satanic pride and the pride of self-glorification. A vain person may have a certain understanding of values and a readiness to conform to their call. In addition, vanity can be restricted to certain spheres of values. For example, one can be proud of one's appearance and yet be indifferent to higher values, such as intelligence or moral integrity (von Hildebrand 2020, p. 471).

The fourth basic type of pride von Hildebrand lists is haughtiness (von Hildebrand 2020, pp. 473–75). Like vanity, haughtiness is partially compatible with a value-response attitude because it does not imply a hatred of values. In this type of pride, values are not used as a means of self-glorification. A haughty person can have respect for certain values and recognize the moral norms that are based on them. In certain cases, she may also recognize legitimate authority. However, such a person is incapable of displaying moral emotions such as contrition, compassion, or gratitude—contrition, because haughtiness does not permit the admission of personal moral failures, even to oneself; compassion, because it is based on the idol of pseudo-virility; and gratitude, because it refuses to recognize one's indebtedness to another person. A haughty person is also averse to admitting the limits of her strength and independence because she sees this as an admission of personal weakness, frailty, or guilt. She recognizes only one evil, which is weakness.

We find this remarkable fourfold categorization of pride both in von Hildebrand's *Ethics* and in the edited materials on the roots of moral evil. The second type of pride, namely, the pride of self-glorification, is elaborated in great detail in the edited materials. There, von Hildebrand makes three particularly noteworthy points.

The first is that the pride of self-glorification is not always based on the values one really possesses, but also on the values one presumes to ascribe to oneself. For example, a conceited person may pride herself on her presumed intelligence, education, or appearance. Second, this pride can even be based on what is a disvalue in reality, for example, when a man prides himself on being macho. Third, von Hildebrand even considers the possibility that human pride is wounded by the realization that one does not have certain values. As in the case of pride of self-glorification based on values that one actually has, the severity of this pride is proportional to the height of the value that the person realizes he or she lacks. If one's pride is hurt by not having, say, a beautiful appearance, it is much more innocuous and superficial than the wounded pride of acknowledging oneself as insufficiently intelligent or artistically skilled.

The passage in which von Hildebrand analyzes the negative attitudes that this type of pride can provoke is very interesting. The first of these is the envy that someone whose pride is hurt by not having a certain value feels toward the one who has it—whether actually or allegedly (von Hildebrand 2023). What is even worse than envy is the second attitude, which is resentment. This attitude is characterized by an effort to overcome the feeling of one's own inferiority by belittling or even denying the superiority of the one to whom one feels inferior. To give an example, Jack hates John because his pride is hurt by the fact that John is much more intelligent than Jack. In order to avoid an unpleasant feeling of inferiority, Jack makes himself believe that intelligence is not actually desirable. The danger of this attitude is that, unlike envy, of which the envious person is often conscious, resentment is usually unconscious and unacknowledged. It is a moral poison that acts slowly and covertly, but its effects are all the worse. Von Hildebrand intriguingly notes that resentment is linked to satanic pride by a hatred of values, and a desire to dethrone them. But unlike satanic pride, the reason for this desire to dethrone a given value is not simply the hatred of the metaphysical power of the given value, but the inability to bear the superiority of the person imbued with that value (von Hildebrand 2023).

As in the case of concupiscence, von Hildebrand does not limit his discussion to a description of the basic types of pride, but makes a number of distinctions within pride

that cut across the types. One of them is the distinction between dynamic and static pride. Static pride is pride that stems from the knowledge that one is a bearer of certain (alleged or actual) values. Dynamic pride is the craving for the acquisition of certain values. While in the pride of self-glorification, we find both of these types of pride, or rather, this type of pride can take both static and dynamic forms, vanity and haughtiness are, by their nature, largely static kinds of pride, von Hildebrand notes.

I would like to conclude this section on a critical note. In his descriptions of the pride of self-glorification, von Hildebrand makes the following three seemingly contradictory assertions: (1) taking pride in merely imagined values is less serious than being proud of the values that one actually possesses;[13] (2) the Pharisee takes pride in merely imagined (non-existent) values;[14] and (3) Pharisaic pride is the morally worse form of the pride of self-glorification.[15] Von Hildebrand does not explain anywhere how these three statements can be simultaneously true. The following is my best guess at how this might be the case: Although (1) is true even in respect of moral and religious values, since the value-response attitude presupposed for the acquisition and possession of these values is incompatible with the attitude of pride, it is impossible for a person to take pride in moral or religious values *if they are real*. For this reason, in the case of moral or religious values, making a comparison between pride taken in real values and pride taken in merely imagined values is pointless. However, if one were to make such a comparison, as a thought experiment, it would reveal, even in this case, that pride taken in real values is worse than pride taken in those that are imagined. This is my best guess at how the three aforementioned statements can all be true at the same time. In any case, it is a shortcoming of von Hildebrand that he does not anticipate this objection and does not attempt to answer it. Since we are discussing unpublished manuscripts, it is possible that we would find this answer in the published version of the text. However, we can only speculate about this possibility. To my knowledge, the unpublished material in von Hildebrand´s *Nachlass*—which is all we have in this case—includes no such text.

**5. Hatred**

The theme of hatred was already mentioned in connection with von Hildebrand's claim that moral evil and moral good are polar opposites. I would like to return to this theme in this final section of the paper. The main reason for this is the fact that hatred, along with concupiscence and pride, is the central theme of the edited materials on the roots of moral evil.[16] Moreover, by presenting von Hildebrand's analysis of the nature, main forms, and causes of hatred, it becomes more obvious why von Hildebrand considers hatred to be the polar opposite of love.

Right at the beginning of his analysis of hatred in the edited materials, von Hildebrand attempts to specify what type of opposition is found between love and hatred. He does so by comparing it to other types of opposites. The contrast between love and hatred is, he submits, fundamentally different from, say, the contrast between joy and sadness, or esteem and contempt. This is because these opposites have their origin in the antithetical character of the objects to which they refer. By contrast, the opposition of love and hatred has its origin in the antithetical character of the basic intention and center in the human person in which it originates. Let me explain. Contempt is a due response to a disvalue; esteem is a due response to a value. Sorrow is a response to a disvalue; joy is a response to a value. Hatred, by contrast, is not an expression of a value-response attitude differing from love, only in that it refers to an object, the importance of which is antithetical to the importance of the object of love. "Hatred in the strict sense", von Hildebrand contends, "is never a value-response attitude. Its venomous dark character is qualitatively the very antithesis of the victorious intrinsic goodness of charity" (von Hildebrand 2023). Thus, unlike the aforementioned opposites, love and hatred come from opposite centers in man. While love comes from the reverent, value-responding center, hatred has its roots in pride, concupiscence, or both.

Since the connection between concupiscence and hatred is less direct and articulate, von Hildebrand primarily elaborates upon the link between hatred and pride. This connection depends on what type of pride and hatred we are speaking about. The worst form of pride, that is, satanic pride, is the actual source of the worst form of hatred, which is the hatred of God (von Hildebrand 2023). The less severe forms of hatred are not so intrinsically linked to pride. What is crucial to the emergence of hatred from these forms of pride is whether it is a pride based on the (actual or alleged) possession of a particular value or a pride wounded by the knowledge that one does not possess a particular value.[17] In the first case, the person will experience hatred only accidentally, that is, only upon discovering that the possession of the qualities one prides oneself on is rivaled or surpassed in someone else. The second case is much worse. Pride that is wounded by the awareness of one's own inferiority often leads to resentment, a condition that, as already mentioned, poisons the human soul and human relationships.

However, there are also types of hatred that are neither rooted in pride nor in concupiscence, von Hildebrand notes. One of them is hatred that is associated with revenge. In von Hildebrand's view, in order to grasp the link between revenge and hatred, we must first understand that revenge is not a primitive predecessor of the principle that crime deserves punishment. The avenger does not want to punish the crime in order to make justice triumph, but to hit back when he has been injured (von Hildebrand 2023). That means that revenge is impossible without believing oneself to have been harmed by the person deemed responsible, whether this is true or not. Second, revenge gives a particular kind of satisfaction to the avenger. This satisfaction arises by causing suffering to the one who (actually or supposedly) wronged him. Part of this satisfaction is connected to a feeling of exacting, so to speak, "payback" for the original injury. And it is precisely hatred that causes this desire and that makes its satisfaction a pleasure. This is, in sum, the relationship between hatred and revenge.

I will conclude my presentation of von Hildebrand's discussion of hatred by alerting the reader to an intriguing response by von Hildebrand to the following question: "How do hatred and other negative feelings arise in the heart of a person when they are rooted neither in pride nor in concupiscence?" His answer is that hatred and other negative feelings are, in this case, a morally decent person's reaction to repeated offenses. In describing the dynamics leading to this abominable result, von Hildebrand develops an intriguing Christian interpretation of Plato's *thymos*, that is, the spirited or irascible part of the human soul (Plato 2003, Book IV). He argues that when someone is unjustly attacked, his or her affective reaction to this attack is not an expression of either pride or concupiscence. It is morally legitimate. Nonetheless, this originally legitimate reaction can (and often does) become the point of origin of hatred of the perpetrator. In explaining how this happens, he draws on the conceptual resources laid down in his magisterial treatment of what he terms cooperative freedom in chapter 25 of *Ethics* (von Hildebrand 2020, pp. 331–53). In this chapter, he explains that there are two ways in which our freedom cooperates with our affective reactions. The first is when we approve of them. He calls this act "sanctioning". The effect of this act is that what was originally just an instinctive affective reaction becomes an affective response that is fully our own. The second is a negative act. He calls it "disavowing". Through this act, we distance our personal, spiritual self from our affective responses.

Von Hildebrand highlights two crucial points regarding this capacity of inner freedom toward our affective responses. The first is that many of us are unaware of this capacity. He calls these people "morally unconscious". To be morally unconscious is to be unaware of our ability to sanction or disavow our affective responses. The second point is the moral relevance of sanctioning and disavowing. To disavow a morally illicit affective response, such as malicious joy, is morally good. In fact, in von Hildebrand's view, it is even morally obligatory because it is not morally licit to live in naïve solidarity with such a response. Now, while we do not have a moral obligation to dissolve such affective responses immediately and directly by will alone (a plainly impossible task in any event), we are morally compelled

to disavow them whenever we become consciously aware of their occurrence; only then can we avoid living in naïve solidarity with *all* of our affective responses. Not only that, but we are also ethically bound to sanction our morally appropriate affective responses.

Applying these insights to the possibility (and, arguably, moral duty) of the cooperation of our will with the affective reactions of our irascible center to repeated offenses, he describes the dynamics leading to the emergence of hatred in the heart of the morally unconscious person. Such a person will experience her natural hostile reaction to the repeated offenses as completely legitimate. In fact, such a reaction *is* legitimate, provided that the insults are real and not just imagined, says von Hildebrand. The slippery slope starts, however, when this initially quite justified indignation of the injured against the one who injures him develops into a full-fledged hatred. This is possible precisely because the injured person is unaware both of its ability and of its moral duty to disavow morally negative affective responses such as a hatred of the offender. "Whereas the morally conscious man will disavow these impulses with his free spiritual center and counteract them, this morally unconscious type will fall prey to the immanent logic of his hostile impulses. [...] He looks at his hatred as something morally legitimate because he has preserved the original consciousness of the legitimacy of his hostility during this entire process of degeneration". (von Hildebrand 2023)

## 6. Conclusions

The aim of this article is to introduce the reader to a remarkable conception of the roots of moral evil found in the published and unpublished works of Dietrich von Hildebrand. A specific feature of this approach is the unusual combination of a highly systematic approach to the problem at hand (an attempt to identify the two basic roots of all forms of moral evil) with the immense breadth of the phenomena described and analyzed. It is as if the Aristotelian sense of the systematic nature of philosophical knowledge here meets the phenomenological sensibility to the originality and mutual non-reducibility of various moral phenomena. Moreover, as I have indicated, von Hildebrand's choice and treatment of certain topics evokes the approach of the existentialists.

Although I have tried to place von Hildebrand's conception, at least in some respects, on the conceptual map defined by the most prominent traditional approaches to the problem of moral evil, it is clear that the aim of studying this conception is not to obtain clues for its inclusion in one or another compartment of systematic or historical positions. The most fruitful way to read von Hildebrand's texts is, as with many of the phenomenology-oriented authors, to learn to make the phenomena described present in one's own rational intuition and to see whether the author's way of describing them is true to their intelligible nature. Thus, the study of von Hildebrand's conception of the roots of moral evil is, like the study of much of his philosophical work, an offer of a *symphilosophein*, in the spirit of Husserl's maxim "To the things themselves!"

The obvious Christian, indeed, specifically Catholic, inspiration at many points in von Hildebrand's ethical writings does not mean that we are methodologically moving from this philosophical/phenomenological level to the plane of moral theology. Although it might sometimes seem so, von Hildebrand's ethics is not Christian, or specifically Catholic, in the sense that it methodologically leaves the plane of what can be stated in purely philosophical/phenomenological terms. Its Christian/Catholic inspiration manifests itself primarily in the idea that the moral excellence of the saints far exceeds the moral virtues of even the noblest pagans. The truth of this claim, von Hildebrand argues, is independent of a belief or disbelief in Christian mysteries. It depends solely on whether it can actually be shown in philosophical terms that specifically Christian virtues such as humility or neighborly love are a higher realization of the ideal of moral excellence than, say, the cardinal virtues of natural ethics, that is, wisdom, courage, moderation, and justice, and whether the Christian saints incorporate these higher virtues and ideals to a higher degree than the "noblest pagans". It is in this sense that von Hildebrand's ethics and *Ethics* are Christian (von Hildebrand 2020, pp. 479–89).[18]

**Funding:** This research was funded by IGA (Palacký University), grant number IGA_CMTF_2023_004.

**Institutional Review Board Statement:** Not applicable.

**Informed Consent Statement:** Not applicable.

**Data Availability Statement:** Not applicable.

**Conflicts of Interest:** The author declares no conflict of interest.

## Notes

1.  Despite being born in Florence, Italy, and living in the US since 1940, Dietrich von Hildebrand's family roots were in Munich. His father, Adolf von Hildebrand, an important artist and art theorist, built a villa for his family located beautifully on the outskirts of the famous *Englisches Garten*, the huge city park in the center of Munich. Dietrich von Hildebrand inherited this villa and lived there with his family until 1933, when, as one of the most outspoken opponents of Nazism, he had to leave Germany. Although, after the war, he never returned to Munich; he decided that his literary bequest should be deposited in the Bavarian State Library, where the bequests of other Munich phenomenologists are also deposited. On von Hildebrand's fight against Nazism, see von Hildebrand (2000, 2014).

2.  Regarding concupiscence, the first main root of moral evil according to von Hildebrand, and the difference between *Ethics* and the edited material, is that while in the former, many different forms of concupiscence are merely listed, in the latter, these same forms are elaborated at length (von Hildebrand 2020, pp. 458–59). Regarding pride, the second main root of moral evil according to von Hildebrand, the new aspect with respect to *Ethics* is, in the edited materials, the discussion of various problems and points arising from the analysis of pride in *Ethics*, but left out from that work. In general, one may say that regarding both concupiscence and pride, the novelty of the edited material with respect to *Ethics* lies mainly in offering explications and further developments of themes mentioned or implied in *Ethics* rather than in presenting something entirely new. The most substantial new idea to be found in the edited material with respect to *Ethics* is the theme of hatred. In *Ethics*, hatred is mentioned as one of the evil responses rooted in pride (von Hildebrand 2020, p. 476). But it is only in the edited materials that it is given proper attention and is analyzed in its various forms and its complex relations not just to pride, but also to concupiscence and revenge.

3.  For the purposes of this study, I take the two terms as being equivalent.

4.  For Plato's account, see Plato (2003, Book IV); for Aristotle, see Aristotle (1985, Book VII). For a more detailed comparison of von Hildebrand's account of acratic action with Plato's and Aristotle´s, see Cajthaml (2019, pp. 91–122).

5.  Note that, for Scheler, the acts of preference (*Vorziehen*) and depreciation (*Nachsetzen*) of values are cognitive acts, not volitional ones.

6.  Scheler's account is arguably more sophisticated than this, but is still unable to adequately explain situations similar to the dilemma of going to the amusing event vs. assisting one's friend. Cf. the critical discussion of P. H. Spader's attempt to justify Scheler's position in the face of Hildebrand's criticism (Cajthaml 2019, pp. 64–69).

7.  In the edited materials, von Hildebrand occasionally uses the expression "*religio* to the world of values", where *religio* is equivalent to a "value-response attitude". The term "religio" is used in these contexts in an explicitly moral sense but with implicit religious connotations. As just mentioned, von Hildebrand uses it as equivalent to the value-response attitude in the sense of the fundamental, ultimate, most foundational attitude of the human person toward the "world of values", particularly, the "morally relevant values". "Religio" hence means the very foundation of all moral goodness in the human person. Simply put, the moral life consists of, according to von Hildebrand, fulfilling the task of giving the "world of values", that is, the entire sphere of what is intrinsically precious, its due and to respond adequately to its "call". From the moral point of view, the most important is to give its due to morally relevant values, that is, values to which to respond makes us good in the specifically moral sense as opposed to, say, values to which to respond makes us good artists or scientists. Since, however, for von Hildebrand, the ultimate source of all values is God understood in the specifically Christian sense, that is, as the Holy Trinity, the attitude of *religio* is implicitly, ultimately, an attitude toward God, and to that extent, it is a "religious attitude". But this attitude is different for a Christian or other believer in a personal God, who serves God consciously and deliberately through his service to the "world of values", and for an atheist who, if he serves the world of values, serves God without realizing it. The same is true in reverse; the believer in God consciously opposes God by rejecting the basic value-response attitude to the world of values, whereas the atheist does not. This answer could, of course, be further refined, but it is perhaps sufficient for a basic understanding of von Hildebrand in this respect.

8.  Clearly, the motivation behind the decision to help a friend in need would require a much more detailed analysis. It is certainly not a response to a single value, but a motivation that may, depending on the actual situation, include a whole host of values in many imaginable mutual constellations. These values would probably include, among others, the value of fidelity to one´s friend, the value of his or her life (if it is threatened), and the value of his or her well-being.

9.  On the basis of this chapter, it is safe to assume that the mental conflict and the possibility of acting against one's better knowledge illustrated by the above example of the friend and the amusing event presupposes the first (and perhaps most prevalent) moral character mentioned, i.e., a morally struggling person in whom the antithetical centers openly struggle with each other. This

shows at the same time that, according to von Hildebrand, the problem of the roots of moral evil lies not primarily at the level of the will, which determines the direction and goal of individual actions, but at the deeper level of basic human attitudes, which, as just indicated, admits several basic constellations between antithetical "centers".

[10] In the Christian tradition, this view dates back at least to Augustine, who, for his part, probably adopted it from Neoplatonism.

[11] This becomes more obvious from von Hildebrand's description of hatred, discussed below.

[12] See above, note 2.

[13] Cf. the following passage: "The man who abuses the possession of authentic values by considering them as mere means to his grandeur and superiority, is more responsible because he is at least able to grasp a real value and, nevertheless, instead of gratitude to God in the case of exterior perfections or of the consciousness of his insufficiency with respect to the tasks to which his capacities are ordered in the case of intellectual values, he turns back to himself and uses these values as means to his grandeur. The conceited man who glorifies himself in imaginary perfection, by contrast, is already disabled by his naïve self-satisfaction from distinguishing the presence of a real value. This pride, though obviously immoral as such, as any pride, above all affects his intelligence and his capacity to grasp reality, and renders him more ridiculous than wicked. The more we are confronted with real values, the more we have really received a gift from God's bounty, the worse and more serious is the attitude of pride" (von Hildebrand 2023).

[14] "The Pharisee, as someone who is completely dominated by pride, can obviously never possess true moral and religious values. He glorifies himself *a fortiori* always in non-existent values, in contradistinction to the man who is proud about intellectual values or exterior perfections" (von Hildebrand 2023).

[15] This third claim follows from the following three assertions that von Hildebrand repeatedly makes: (1) the higher the values that one takes pride in, the morally worse the pride itself is; (2) the highest personal values are moral and religious; (3) Pharisaic pride seeks glory in alleged moral and religious values.

[16] Cf. above, note 2.

[17] This distinction was already mentioned in the discussion of pride.

[18] At the time this article was ready to print, the edition of the materials from von Hildebrand´s *Nachlass* I cite in it was not yet paginated. There is not much point in quoting these materials according to the signatures in von Hildebrand´s *Nachlass*. Therefore, I exceptionally quote this source without citing the pages.

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
