# Peer review of "Von Hildebrand on the Roots of Moral Evil"

_religions, doi:10.3390/rel14070843_

Round 1

Reviewer 1 Report

This paper takes up the task of elucidating Dietrich von Hildebrand’s account of the roots and nature of moral evil. The paper surveys von Hildebrand’s work – both previously published works and the forthcoming posthumous work — to detail the typologies von Hildebrand uses to make sense of moral wrongdoing and to put his ideas into conversation with classical ideas about akrasia and the privatio boni. The writing style is clear and the distinctions between different concepts provides a map for charting different kinds of wickedness. The discussion of hatred, in particular, is strong and interesting.

Here are a few small suggestions for revisions:

1(1)     The abstract is largely a copy-paste of the introduction and should be its own piece of text which more succinctly surveys the argument of the paper.

2(2)    What does “value-response attitude” mean for von Hildebrand, and why is “religio” an apparent synonym? Explaining what von Hildebrand means by “religio” would clarify this argument and, possibly, make this paper more evidently connected to the topic of religion.

3(3)    The block quotation that runs from line 577 to line 601 is too long and is not interpreted. This could be substantially cut down and more commentary could be provided.

One more substantive recommendation is to justify why readers might be interested in von Hildebrand’s account of the roots of moral evil. In the introductory section, the primary justifications given for this elucidation are that (1) no one has written about this topic before in scholarly literature, (2) this paper draws on primary resources that have not yet been engaged with, and (3) these soon-to-be-published resources bring a “new, richer perspective on the theme.” The first two of these three can be part of a justification for a published article, but are not in themselves sufficient. After all, there are several topics that no one has written on because they are not especially interesting or revealing, so a lacuna in scholarly discourse is not motivation enough, and readers who are interested in von Hildebrand on the roots of evil can presumably consult the book itself when it hits shelves.

The third of these three, however—that the view of moral evil presented in von Hildebrand’s Ethics and developed elsewhere in the Nachlass—is more compelling, and the author would be well-served to make more explicit what value this “richer perspective” has for moral philosophy, the history of phenomenology, and/or Christian ethics. In short, why turn to von Hildebrand on this question? What problems can his account of the roots of moral evil help us resolve, or ask in a new way? Especially since so much of this work is a matter of clarifying categories (the four kinds of pride, e.g.), readers will want an indication as to why we might find von Hildebrand’s conceptual framework valuable for making sense of moral phenomena. We get glimpses of this (for instance, the sentence at the end of part III, “This remark is worth of careful reflection in today’s world dominated by work, efficiency, and pragmatism as an existential attitude”), but an introduction (before part II) that outlines why von Hildebrand’s view of moral evil deserves further consideration would dramatically elevate this piece and give it significance that goes beyond a teaser for the forthcoming volume. This is especially the case since the series is not dedicated to the work of von Hildebrand specifically but is aimed at readers interested in Continental Philosophy and Christian Beliefs.

Author Response

1) I agree with the suggestion. I have rewritten the absract accordingly.

2) I have expanded note 7, offering some clarification of what "religio" means in vH´s texts on the roots of moral evil. I have inserted a new section (lines 139-177) where I tried to explain (in a succint way) what "value response attitude" means in the given context. It is a broad topic. A full response would require an analysis of vH´s different writings on the topic starting with Sittlichkeit und ethische Werterkenntnis till at least his (Christian) Ethics. This is clearly impossible in this paper. So I tried to be brief. However, I agree that explaining this aspect of vH´s though is crucial for the topic of my paper. Originally, I tried to avoid it because of space constraints and the considerations regarding the flow of the exposition. But I recognize from your reaction that some explanation of "value-responding attitude" is necessary.

3) I abbreviated the quotation substantially and turned some of it into a paraphrase. I finally decided to keep this point short because I added text elsewhere and did not think to be a good idea to expand the paper too much. 

Regarding the more substantive recommendation: I agree that the scientific importance of an article must include more than just the proof (or an argument) that the topic has not been dealt with so far. But I think that in this case the best way how to justify writing on the topic is to present von Hildebrand´s ideas as clearly and faithfully as possible. After all, we speak about the roots of moral evil, a topic which is uncontroversially important both existentially and theoretically. And since von Hildebrand´s treatment is unique both in its scope and its analytical vigor, I think no further justification of why writing on this subject is necessary. I personally do not see any comparable account of the roots of moral evil in terms of deapth and scope not just in Husserl, Heidegger, Scheler or other phenomenologists but also in classical thinkers as Plato, Aristotle, Agustine or Aquinas. This being said, I prefer to show the importance of the topic of my paper by presenting von Hildebrand´s account as claerly and faithfully as I can rather than making strong introductory statements. With this I do not mean that your recommendation is not to the point. Someone might use it profitably. But for me it would mean to go against the grain. Hence I would like to take the liberty of keeping with my approach although I see that other might contest it with good reasons. 

Reviewer 2 Report

The length of the paper is standard, usual for scientific papers. The topic is borderline, interdisciplinary, philosophical and partly theological. Due to the focus of the contribution, the paper is publishable in the journal Religions. The topic dedicated to the author von Hildebrand is not one of the frequently discussed topics. Moral evil is a topic that has been dealt with by many philosophers, e.g. also Aurelius Augustinus. Among other things, the paper looks for points of contact between Hildebrand's understanding of moral evil and selected Western philosophers. The author is fully aware of Hildebrand's phenomenological basis. Hildebrand himself disagrees with the statement of Plato's Socrates, who says that man does not do evil consciously. Sometimes a person chooses the more accessible, more pleasant option, notes Hildebrand. The author finds certain correlations in the sense of looking for parallels in the contradiction in the soul in Plato - in his triadic conception of the soul. However, Hindelbrand is original in this, the author states. Hildebrand certainly does not perceive evil, like Plotinus, as a deprivation of good. Hildebrand argues that lack of love is indifference but not hatred. He considers pride and lust to be the source of evil. The author tries to argue with Hildebrand. I do not take a position on the polemic, it is not the job of the reviewer to argue with the author. The metaphysical status of negative sources remains open for Hildebrand. Hildebrand therefore does not know this cause and cannot justify it metaphysically, he considers this question unanswerable. Hildebrand divides lust and pride into subspecies. In this context, he also categorizes people according to their natures. He also discusses sadism as a particularly perverted desire for pleasure. It also touches on the issue of avoiding pain. According to Hildebrand, desirability also manifests itself in the desire for distraction. It also deals with laziness. Even an active person can be spiritually lazy, the author draws attention to this idea of Hildebrand. Hildebrand claims that some people fall into illusions, they do not perceive reality. This is how they try to get rid of something depressing. It also manifests itself in the search for superficial entertainment. The author also processes Hildebrand's understanding of pride. Hildebrand connects it with the aspiration to metaphysical sublimity. He also recognizes the pride of self-glorification, vanity and arrogance. In conclusion, Hildebrand talks about hatred. He considers hatred of God to be the worst form of hatred. He also describes the development of the middle part of the thymos soul from Plato in the intentions of Hildebrand's anthropology. The conclusion is sufficiently long and concise. The bibliographical references are almost exclusively oriented towards Hildebrand. There are also relatively few of them.

I think that the submitted paper does not reach the quality to be published in a quarantined journal. I justify this by the fact that, in the vast majority, it is only an interpretation of Hildebrand's thoughts and opinion positions. It is done thoroughly, but a contribution in a quarantined journal should bring several innovations. I recommend the author to publish it in a less prestigious scientific journal, or to radically revise it and build its concept on the contribution of some new ideas.

Author Response

The reviewer suggests that the paper be submitted to less prestigious journal or reworked substantially because, in its present form, it does not contain enough "innovations" but is prevalently only a intepretation of vH´s ideas.

I admit that the aim of the article is primarily to present, contextualize, and comment on vH´s account of moral evil. It is such on purpose, not by accident. The crucial question then is whether such an article is sufficiently "innovative" for a Q1 scientific journal. My response is that there are different types of "innovations." In my paper, I do not develop a new original philosophical theory or argument. If that were the only relevant meaning of "innovation" my paper would be disqualified. But if "innovation" is construed as "bringing into a contemporary discourse in the given field new important ideas" then I submit that my paper is sufficiently "innovative". Here are the reasons for my claim:

On the account of many experts, phenomenology is presently undergoing several new developments. One of them is the increasing awarness among specialists today that the early phenomenology, to which vH belongs, was for decades undeservedly marginalized. Rediscovery of the major contributions of these early phenomenologists is thus a topical task, not just for historical reasons, but also for genuinly philosophical reasons. Now vH was one of the major figures of this phase of the phenomenological movement. He was not just a student of Husserl. Among other things documented in the articles of Karl Schuhman, the historian of the early phenomenological movement, he was also one of the founders of the Jahrbuch für Philosophie und phänomenologische Forschung and one of the first contributors to this Jahrbuch with his works Idee der sittlichen Handlung and Sittlichkeit und ethische Werterkenntnis. 

Once the importance of vH is conceeded, the next step is to realize that his main achievements were in the field of moral philosophy and that his magnum opus in this field was (Christian) Ethics. For this reason, my discovery of so far unkown almost 400 pages of manuscripts and some 200 pages of transcripts in vH´s Nachlass dealing with the roots of moral evil (the main topic of part four of vH´s Ethics) is so far an unparalleled found in the history of Hildebrand scholarship. My paper presents these newly discovered ideas at the moment, when the critical edition of the discovered material is just about to be published. So if anything is topical, I think this article is. Whether my discovery will have a significant impact on the studies in early phenomenology remains to be seen. But the potential is demonstrably there. On the basis of this argument, I consider my article sufficiently innovative in the context of the given field of research and in the context of a special issue of the journal on continental philosophy.

Reviewer 3 Report

The abstract is too long, too detailed.  It should be shortened.

Even though I realize that this article is meant to be mostly expository, it seems as if there are places where the author should have taken a more analytic/critical stance toward von Hildebrand's account of moral evil.  So, for instance, the discussion of the ontological status of the opposing "centers" in human nature (lines 187ff) seems unsatisfying because 1) von Hildebrand makes certain metaphysical assumptions (e.g., there is a God who cannot be the source of evil) and 2) it seems as if one could be critical of von Hildebrand for not answering the ontological question regarding the final cause of all evil (why is it unanswerable on the philosophical level?).  In addition, I found myself asking questions of von Hildebrand's account to which there are no answers in the article.  Is it ever (morally) legitimate to be proud?  Is the "elegant man" (lines 358ff) a morally bad person?  Is the hatred associated with revenge (lines 530-542) morally justified, according to von Hildebrand, even though it is not rooted in either pride or concupiscence?  Answering these sorts of questions would go a long way toward clarifying von Hildebrand's account.

Though there is some attempt to situate von Hildebrand's account in the context of both Christian religious thought and the Western philosophical tradition, it seems as if more could be done in this regard.  So, for instance, the strong Christian influence on von Hildebrand's ethics is noted only in passing, though that seems to be important if only because it might call into question the extent to which his account is phenomenological.  It seems that von Hildebrand's ethics is largely a (Christian) ethics of virtue, yes?  How does it compare with other ethics of virtue in the Western philosophical tradition?

Author Response

The first first comment of the reviewer was that I should take a more analytical/critical stance toward von Hildebrand´s ideas. I grant this point. In the new version, I included lines 526-546. I express there what I take to be the most serious immanent criticism to one particular aspect of von Hildebrand´s account of moral evil. In general, I criticize vH where I see it appropriate (there are numerous such points in my book on his moral philosophy). In the area covered by the reviewed paper however I do not find such problematic issues. In particular, I do not find the claim that (Christian) God can be no cause of any evil as controversary. If someone finds it such, he is free to take issue with vH, Plato, Augustine or Aquinas regarding this matter. Similarly, I agree with von Hildebrand´s view that the ultimate cause of evil is veiled by a mystery. Seen in Christian perspective, why would a creature who has no innate tendency or propensity to evil--since it was created all good--turn aginst its Creator? Christians believe that some pure spirits ("fallen angels") "once" did. Had something like this happend, this would be beyond human capcity of rational understanding, says the tradition. I agree with that. However, I would be interested to see arguments contesting this approach and I would certainly look at them carefully.

To reviewer´s  questions he did not find answered in the article:

1) Is it ever morally legitimate to be proud (according to vH)? I think the response is obvious from vH´s understanding of pride expressed in the article. If pride is the root of evil every form of pride is by defintion evil. Obviously, the second question is whether vH is right in thinking that every form of pride is morally bad. Without trying to settle this issue definitivelly, I would like to point out the fact that while English covers morally legitimate and morally illegitimate forms of pride by one word "pride", German  distinguishes between morally negative form of pride (Hochmut) and morally neutral, or at times even morally positive (Stolz). To give an example, I can say in English "I am proud of my accomplishments." The equivalent sentence in German would be "Ich bin stolz auf meine Leistungen." Note further that when vH writes about pride in German (German is his first philosophical language), he always uses the word "Hochmut". So when he writes about "pride" in English, he in fact has in mind Hochmut in German. 

2) Is the elegant man a morally bad person? This is a good question. The simple answer implied in what is written in the paper is that he is to the extent that he is proud and concupiscent (vH details elements of pride and concupiscence in this moral type). But, according to vH, the elegant man is certainly not a person totally possessed by pride or by concupiscence. So there could be many morally good traits in such a man.

3) Is the hatred associated with revenge morally justified? From von Hildebrand´s standpoint it follows that no hatred is morally justified. Hence also hatred associated with revange is morally bad affective response. However, he says, we must distinguish between the initial natural reaction of a person to an offense which is morally legitimate and an outspoken hatred which can develop from this reaction which is, as any hatred, morally bad. As mentioned in the paper, the degeneration of the original affective reaction into full-fledged hatred is the result of the lack of moral consciousness: because the person is not aware of her capacity (and in fact moral duty) to sanction morally positive affective responses and disavow the morally bad responses she fails to undercut the development of her hate by disavowing with her cooperative freedom the morally negative affective reaction.

I agree that more could be done with respect of situating vH´s account in the tradition of Christian religious thought and Western philosophy. Given the space constraints, I decided to include at least the lines 174-180. I indicate there the relationship between vH´s and Aristotle´s acount of moral virtue with a reference to a secondary literature where these two accounts of moral virtue are compared in detail. I have more material to insert regarding the relationship between vH´s ethics and contemporary virtue ethics but it does not concern moral evil but motivation of moral action, conceptualization of the good as an object of desire vs. value as the intrinsic preciousness of a being, etc. However, I prefer to publish this material in a fully developed form in separate articles. These issues are quite complex and in my view their insertion into this paper would distract the reader from the main topic which is vH´ account of moral evil.   

Round 2

Reviewer 2 Report

New paragraphs were added to the work. It is a positive change. Even so, I am not convinced that this is a study that has acquired new qualities through revision. I don't think it's a comprehensive overhaul. It was not significantly reflected in the bibliography either.

Author Response

I appreciate your assessment of my paper. It seems that we do not have an agreement on a basic point. Obviously without considering it perfect I nevertheless think that this paper is in its present form worthy of being published in Q1 journal. You contest it. I gave my reasons for my claim in the response to your first comments. In your response to my response you do not engage with my arguments but restate your previous conclusion. I appreciate it but I do not see how our discussion could move further. In any case, thank you very much for taking the time to read and evaluate my paper.